# GEOMETRIC ALGEBRA ATTENTION NETWORKS FOR SMALL POINT CLOUDS

## ABSTRACT

Much of the success of deep learning is drawn from building architectures that properly respect underlying symmetry and structure in the data on which they operate—a set of considerations that have been united under the banner of geometric deep learning. Often problems in the physical sciences deal with relatively small sets of points in two- or three-dimensional space wherein translation, rotation, and permutation equivariance are important or even vital for models to be useful in practice. In this work, we present rotation- and permutation-equivariant architectures for deep learning on these small point clouds, composed of a set of products of terms from the geometric algebra and reductions over those products using an attention mechanism. The geometric algebra provides valuable mathematical structure by which to combine vector, scalar, and other types of geometric inputs in a systematic way to account for rotation invariance or covariance, while attention yields a powerful way to impose permutation equivariance. We demonstrate the usefulness of these architectures by training models to solve sample problems relevant to physics, chemistry, and biology.

## BACKGROUND

Deep learning has been immensely successful in solving a wide range of problems over the last several years, driven in large part by identifying appropriate ways to embed structure of data and symmetry of problems directly into the architecture of the network—an idea at the core of geometric deep learning (Bronstein et al., 2021). Some applications of geometric deep learning include the use of convolutional filters in CNNs to attain translational equivariance, or graph convolutions in graph neural networks for permutation equivariance.[1] Building symmetry into the architecture of a deep neural network can improve the network's data efficiency and guarantee important analytical properties without having to rely on the model to learn to approximate them from training data.

In this work, we derive a family of architectures that is useful in applications from physics to biology, where problems often deal with relatively small point clouds of labeled coordinates. These could be local environments of particles assembling into a crystal (Dshemuchadse et al., 2021), atoms in a molecule interacting with other atoms (Chmiela et al., 2017), or coarse-grained beads representing parts of a protein (Marrink & Tieleman, 2013). In many of these applications without the influence of an external field, we are interested in modeling attributes of the system—such as the identity of a particle's local self-assembly environment, or the potential energy of a group of atoms—which are invariant with respect to rotation of the input coordinates, as well as permutation in the ordering of points. As shown in Figure 1, here we attain rotation invariance by constructing functions from rotation-invariant attributes of geometric products of input vectors from geometric algebra, and permutation invariance by using an attention mechanism to reduce the set of vector products.

---

[1]In this work, we use the following terms to discuss symmetry of functions $f$ and operations $\rho$: $f$ is *invariant* to $\rho$ if it does not change when $\rho$ changes: $f \circ \rho = f$. If $f$ and $\rho$ commute, then we say that $f$ is *covariant* with respect to the operation of $\rho$: $f \circ \rho = \rho \circ f$ (some sources call this equivariance or same-equivariance; the typical definition of equivariance is more general, but we will only discuss $f$ and $\rho$ that are endomorphisms in the context of covariance). Here we use *equivariance* to broadly mean considerations of covariance as well as invariance (since scalars of interest are typically invariant to translation and rotation in physical applications) for simplicity of discussion.

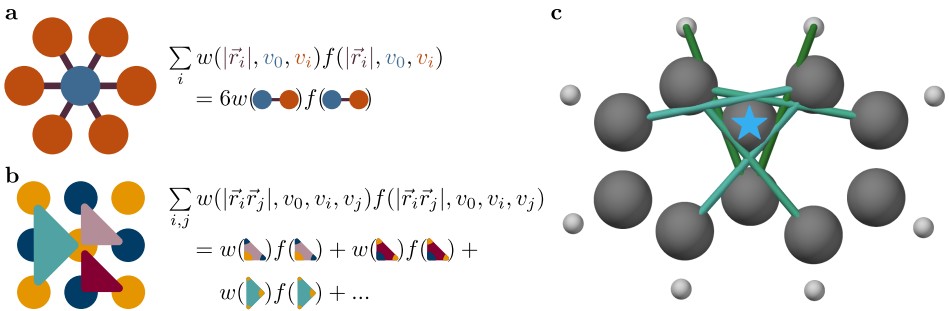

Figure 1: Overall strategy for incorporating rotation and permutation equivariance into deep neural networks using attention mechanisms and geometric products. (a) In the simplest form, our proposed structure uses an attention mechanism over the bond lengths of a cloud of points, each of which carries a value as commonly used in graph neural networks. (b) Geometric products can be used to summarize pairs, triplets, or larger tuples of vectors in a systematic and geometrically meaningful way. Rotation equivariance can be attained by producing rotation-invariant or -covariant quantities in each layer of a model. Here, attention over pairs of bonds—represented by two-dimensional tiles— is used. (c) An attention mechanism reduces the set of generated geometric products to enforce permutation equivariance; the learned attention maps can provide insights into how models operate. In this case, a carbon atom in a naphthalene molecule directs its focus broadly around the aromatic rings in which it is situated, rather than focusing exclusively on its nearest neighbors in the molecular graph. Lighter-colored bonds indicate a greater attention weight for the two atoms sharing the bond with respect to the starred atom.

## RELATED WORK

**"Large" point clouds.** Point clouds are a ubiquitous data structure found in many domains, even outside of the physical sciences. For the purposes of this work, we focus on relatively small sets of points where each point is more information-rich—for example, carrying information about atom identities, local environments, or other attributes—in contrast to those commonly found in computer vision and robotics which may represent the geometry of a mesh or be sampled from an object, but otherwise not have as much information content. We refer to Guo et al. (2020) for a survey of this field, but a few of the recently-developed notable approaches include PointNet (Qi et al., 2017), deep sets (Zaheer et al., 2017), and kernel point convolutions (Thomas et al., 2019).

**Geometric approaches for small point clouds.** Many architectures have been proposed to incorporate rotation equivariance by augmenting graph neural networks with geometric attributes that are known to be rotation-invariant, such as bond lengths and angles. SchNet (Schütt et al., 2017) learns distance-based convolution filters which are used to propagate signals over graphs. PhysNet (Unke & Meuwly, 2019) also refines node representations based on bond lengths, while incorporating a learnable attention mechanism. DimeNet (Klicpera et al., 2019) extends the information used to calculate node-level representations to include angles between bonds. GemNets (Klicpera et al., 2021) are related to DimeNets, and incorporate angle information on quadruplets of points within a graph message passing scheme using an architecture optimized for molecular datasets. GNNFF (Park et al., 2021) generates rotation-covariant results by computing a weighted sum of modulated input vectors based on a graph message passing scheme.

**Group representation-based approaches.** These methods take advantage of group representation theory by first transforming inputs into a space in which rotation-equivariant maps are more easily expressed. This set of methods is powerful, having been used in the past to design rotation- and permutation-equivariant models (Kondor, 2018; Thomas et al., 2018; Anderson et al., 2019), and have even been expanded recently for arbitrary groups (Finzi et al., 2021). Attention-based models have also been utilized in this area: SE(3) Transformers (Fuchs et al., 2020) extend tensor field networks (Thomas et al., 2018) with a self-attention mechanism for increased expressivity by incorporating value- and geometry-dependent attention weights.

**Multivector-valued networks.** The approach described herein applies geometric algebra to train deep learning models on point clouds, but using geometric algebra (also known as Clifford algebra)

to structure the operations of neural networks is not a new concept. Prior work has introduced architectures which generate multivectors using the mathematical structure of geometric algebra as an extension of complex numbers (Pearson & Bisset, 1992; 1994) and directly for geometric applications (Bayro-Corrochano et al., 1996; Buchholz & Sommer, 2008; Melnyk et al., 2020).

The approach we present here is similar to many of the ideas presented above; however, rather than specifying particular rotation-invariant quantities to utilize or learning maps that operate on irreducible representations, we leverage the structure provided by geometric algebra to calculate rotation-invariant and -covariant quantities are of interest. Finally, we use an attention mechanism to attain permutation equivariance in a flexible manner.

## DOMAIN-SPECIFIC APPLICATIONS

We use three applications, spanning scientific domains from basic physics to biology, to motivate the development of equivariant models.

**Crystal structure identification.** From atoms to colloidal particles, matter often organizes itself into ordered two- or three-dimensional structures. One of the core ideas of materials science is that structure is one of the key determining factors for material behavior. With this perspective in mind, when studying computational models of self-assembling systems we often first identify what structures, if any, have formed in our simulations—a task complicated by naturally-occurring thermal noise, crystallographic defects, dataset scale, and potentially the complexity of the structures themselves. Early efforts to automatically characterize structure led to the widely-used Steinhardt order parameters (Steinhardt et al., 1983; Mickel et al., 2013; Boattini et al., 2019), which are rotationally-invariant magnitudes of sums of spherical harmonics over local particle environments. While Steinhardt order parameters can be effective when studying phase transitions or distinguishing among a small number of phases, determining appropriate hyperparameters—including neighborhood size to consider, spherical harmonic order $\ell$ to use, and thresholds to identify behaviors of interest—can be difficult (Mickel et al., 2013). For this reason, data-driven approaches to analyzing structure have been the subject of great interest in recent years (Clegg, 2021). Here we identify local environments extracted from ordered structures using a rotation-invariant classifier built on our attention mechanism. While this task is fairly straightforward to perform based on geometry for single-component systems, suitable methods to apply Steinhardt order parameters to systems with multiple types of particles are less obvious; our deep learning architecture more naturally solves this problem using signals associated with each particle type for every bond.

**Molecular force regression.** One of the most dramatic contributions of deep learning to the field of chemistry lies in constructing fast, accurate approximations of expensive physical calculations (Unke et al., 2021; Behler, 2021). Machine learning models can be many orders of magnitude faster than the methods used to generate their training data, which can bring vastly more detailed and longer-time simulations into the realm of possibility. Central to the applicability of these methods are issues of symmetry and equivariance: any imperfection in rotational invariance of a learned potential energy function could ruin the thermodynamic behavior of a system, so care must be taken in model design to ensure physical behavior. Here we train models using our attention mechanism to predict the atomic forces calculated using *ab initio* molecular dynamics and density functional theory (Chmiela et al., 2017). These models are conservative, permutation-invariant, and rotation-equivariant by construction—all crucial attributes when using models in simulation.

**Backmapping of coarse-graining operators.** When simulating large molecules—such as proteins or other polymers—it is common to employ *coarse graining*: a process by which groups of particles are merged into (fewer) distinct beads, enabling faster simulations by decreasing the number of degrees of freedom of the model (Marrink & Tieleman, 2013). Although data-driven approaches have been highly successful to formulate coarse graining operations in the forward direction (that is, from more-detailed to less-detailed systems), some problems are best solved using the original, fine-grained system coordinates, which are not directly available in coarse-grained simulations. To demonstrate the potential for our geometric algebra attention mechanism on this task, we train models to predict the coordinates of the heavy atoms that form an amino acid from the centers of mass of the nearest-neighbor amino acids in protein entries found within the Protein Data Bank (Berman et al., 2000). Machine learning approaches in the past have treated this problem using simple multilayer perceptrons (An & Deshmukh, 2020) or by encoding the geometry of the problem as

image-type data and learning an image translation process (Li et al., 2020). We treat this problem much like language translation models using transformers (Vaswani et al., 2017), by "translating" the point cloud of the coarse-grained neighboring amino acid coordinates into a set of constituent atom coordinates for a central amino acid.

## METHODS

In this work we formulate deep neural networks using learnable functions consisting of two main parts: (1) a set of geometric products (from the geometric algebra in three spatial dimensions) of input vectors; and (2) a permutation-equivariant reduction over these products using an attention mechanism. We describe each of these aspects below.

### ATTENTION FROM GEOMETRIC PRODUCTS

We begin with a description of the bare essentials of geometric algebra used in this work. For more details, we refer the reader to Appendix A. Briefly, geometric algebra provides mathematical structure to deal with geometric objects—such as points and planes—using a common language for arbitrary numbers of spatial dimensions. The primary objects dealt with using geometric algebra are *multivectors*, which consist of linear combinations of basis elements; in three dimensions, these basis elements are one scalar component, three vector components, three bivector components, and one trivector component. We can calculate the so-called geometric product of two multivectors to yield a new multivector; for example, the geometric product of two vectors yields a scalar (that is the dot product of the vectors) plus a bivector (related to the cross product of the vectors). The geometric product is not commutative; for two general multivectors $A$ and $B$, we denote the product $C$ as $C = AB$. In this work, we use the geometric product to combine groups of input vectors and systematically extract rotation-invariant quantities of interest—such as distances, bond angles, and volumes—for use in network layers.

We specify a computational complexity (i.e. whether to compose calculations from pairs, triplets, quadruplets, or larger groups of points) and, for input point clouds with $N$ points, we construct all $N^2$ possible pairs, $N^3$ triplets, and so on;[2] we call these groups of points a *tuple* here. In addition to a coordinate $\vec{r}_i$, we associate a set of values $v_i$ to each point indexed by $i$ in some space with a given *working dimension* (we avoid calling these vectors to decrease the confusion with geometric vectors; values correspond to the non-geometric attributes of the point, such as type embeddings for atomic systems). To create permutation-covariant functions (producing a value for each input point of a point cloud) or permutation-invariant functions (producing a single output value for the entire cloud), we make use of a simple attention mechanism based on the rotation-invariant attributes of each tuple. Attention has been used widely in applying deep learning to a range of problem domains over the last few years, with particular success in the field of natural language processing (Vaswani et al., 2017). Instead of the dot product self-attention commonly used in language models—which operates on pairs of elements—we use a geometrically-informed attention mechanism that operates on each tuple. We specify four functions: a value-generating function $\mathcal{V}$, a tuple value-merging function $\mathcal{M}$, a joining function that summarizes the geometry and tuple value representations $\mathcal{J}$, and a score-generating function $\mathcal{S}$. The functions have the following uses within the network:

- $\mathcal{V}$ produces features in the working dimension of the model from the rotation-invariant geometric quantities associated with each tuple—specifically, rotation-invariant attributes of each geometric product $p_{ijk...} = \vec{r}_i \vec{r}_j \vec{r}_k...$, which include bond distances, angles, volumes, and other geometric attributes. Geometric algebra provides a framework to systematically extract these attributes: for a general multivector, the rotation-invariant quantities are the scalar component, trivector component, the norm of the vector component, and the norm of the bivector component. In the networks presented here, we use MLPs with a single hidden layer for $\mathcal{V}$.

---

[2]To avoid polynomial scaling, the set of products could be reduced according to the edges of a specified graph, the Voronoi diagram of the point cloud, or by randomly sampling tuples of points; however, we leave these extensions to future work.

- $\mathcal{M}$ merges the 1, 2, 3, or more values associated with a tuple of input points into the working dimension of the model. The form of $\mathcal{M}$ could be a complex learned function, a linear projection for each tuple position, or simply the summation of input tuple values.

- $\mathcal{J}$ joins the rotation-invariant representations from $\mathcal{V}$ and the tuple value representations from $\mathcal{M}$. Like $\mathcal{M}$, it could vary in complexity from a simple sum to a complex learned nonlinear function.

- $\mathcal{S}$ generates score logits from the representation of each tuple, incorporating both geometry and value signals associated with points in the tuple. The results from $\mathcal{S}$, passed through a softmax function, will yield the weights for the attention mechanism. We use MLPs with a single hidden layer for $\mathcal{S}$ in the networks presented here.

We first calculate the multivector geometric products $p_{ijk...}$ of all combinations of input vectors indexed by $i$, $j$, $k$, and so on, up to the specified rank (for example, pairwise attention would produce a two-dimensional matrix of products $p_{ij}$). We then use $\mathcal{V}$, $\mathcal{M}$, $\mathcal{J}$, and $\mathcal{S}$—together with a function extracting the rotation-invariant attributes of a geometric product (which are the scalar component, trivector component, and the norms of the vector and bivector components, depending on how many input vectors are joined *via* the geometric product) into $q_{ijk...}$—as follows for a network producing permutation-covariant outputs $y_i$ for each input point:

$$
\begin{aligned}
p_{ijk...} &= \vec{r}_i \vec{r}_j \vec{r}_k ... \\
q_{ijk...} &= \text{invariants}(p_{ijk...}) \\
v_{ijk...} &= \mathcal{J}(\mathcal{V}(q_{ijk...}), \mathcal{M}(v_i, v_j, v_k, ...)) \\
w_{ijk...} &= \underset{jk...}{\text{softmax}}(\mathcal{S}(v_{ijk...})) \\
y_i &= \sum_{jk...} w_{ijk...} v_{ijk...}
\end{aligned}
\tag{1}
$$

If a permutation-invariant reduction is desired, then the softmax and final sum can be performed over all tuples of the point cloud simultaneously. While $\mathcal{J}$, $\mathcal{V}$, and $\mathcal{M}$ could in principle be used to change the working dimension of the network by stacking permutation-covariant layers, in this work we keep the dimension constant for the sake of easily adding residual connections.

If rotation-covariant (rather than rotation-invariant) behavior is needed for the output of the network, the same attention structure can be used with slight modifications; here, we extract a vector from a learned linear combination of the input vectors $\vec{r}_i, \vec{r}_j, \vec{r}_k...$ and the geometric product $p_{ijk...}$ (which consists of directly taking the vector component from products of odd numbers of input vectors, or multiplying a bivector by the unit trivector to produce a vector in the case of even numbers of input vectors). These vectors can be combined with learned vector weights $\alpha_n$, a scalar rescaling each vector (generated by a learned function $\mathcal{R}$), and the attention mechanism to yield

$$
\vec{r}_i' = \sum_{jk...} w_{ijk...} \mathcal{R}(v_{ijk...}) \left( \alpha_0 \text{vector}(p_{ijk...}) + \alpha_1 \vec{r}_i + \alpha_2 \vec{r}_j + \alpha_3 \vec{r}_k + ... \right).
\tag{2}
$$

We use $\mathcal{R}$ to provide a value- and geometry-dependent rescaling of the output vector, making it easier to generate vectors that lie outside of the convex hull of the input point cloud. Because $w_{ijk...}$, $v_{ijk...}$, and $\alpha_n$ are rotation-invariant and the geometric algebra is coordinate-free, the resulting output $\vec{r}_i'$ is also rotation-covariant; more details of this are shown in Appendix A.

### MODEL ARCHITECTURES

We demonstrate the utility of our geometric algebra attention scheme by training deep networks to solve three problems appearing in physics, chemistry, and biology. For simplicity, all of the geometric algebra attention models presented here utilize pairwise attention with a working dimension of 32 units. Value functions $\mathcal{V}$, score functions $\mathcal{S}$, and rescaling functions $\mathcal{R}$ are simple multilayer perceptrons with a hidden width of 64 units, with layer normalization applied to the hidden layer of $\mathcal{V}$. The network for crystal structure identification uses the mean function for merge functions

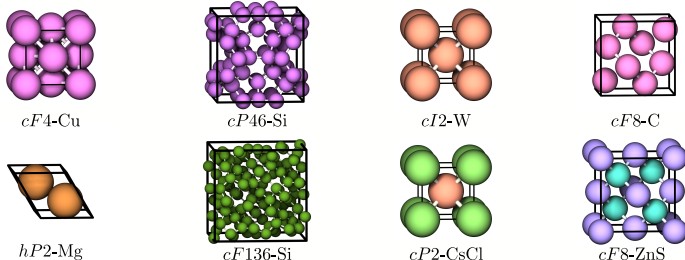

Figure 2: Unit cells of crystal structure prototypes chosen for the structure identification benchmark. Simple and complex structure types—including two binary structures—are included.

$\mathcal{M}$ and join functions $\mathcal{J}$, while the other two applications use learned linear projections. More thorough descriptions of the model architectures—including Python code under the MIT license—are detailed in Appendix B and the Supplementary Information.

**Crystal structure identification.** We train networks using the 12 nearest neighboring particles $j$ around a central particle $i$ extracted from a structure. We use the neighbor bond coordinate differences $\vec{r}_{ij}$ and a symmetrized one-hot embedding vector $[t_i - t_j, t_i + t_j]$ to encode particle type information for each bond. After projecting these per-bond signals up to 32 dimensions, we perform two steps of permutation-covariant attention (using Equation 1) before a permutation-invariant reduction over the point cloud and final projection down to class logits.

**Molecular force regression.** To attain translation invariance, we first convert the set of atomic coordinates $\vec{r}_i$ and one-hot type representations $t_i$ into the atom-centered pairwise coordinate difference matrices $\vec{r}_j - \vec{r}_i$ and symmetrized one-hot type vectors $[t_i - t_j, t_i + t_j]$. After projecting the value signals up to 32 dimensions, we perform six steps of permutation-covariant attention (using Equation 1) before a final permutation-invariant reduction and projection down to a per-atom energy. These energies are summed and we calculate the gradient of the per-molecule energy with respect to the input coordinates to ensure that a conservative force field is learned. We train models by matching the predicted forces to those found in the training data *via* a mean squared error loss.

**Backmapping of coarse-graining operators.** We first apply two layers of permutation-covariant attention (using Equation 1) to the center-of-mass coordinates and identities of amino acids surrounding a central residue, to incorporate local geometry information into the coarse-grained bead representations. We then apply a *translation* layer to convert coarse-grained amino acid coordinates to atomic coordinates, which produces vectors according to Equation 2 by augmenting the tuple representation $v_{ijk...}$ of Equation 1 with labels corresponding to the identity of the atom that should be produced, so that the value is calculated as

$$v_{\text{atom},ijk...} = \mathcal{J}(v_{\text{atom}}, \mathcal{V}(q_{ijk...}), \mathcal{M}(v_i, v_j, v_k, ...)). \tag{3}$$

Following this layer, two rotation-covariant layers (using Equation 2) are applied to the atom-centered, fine-grained atomic coordinates to further refine them.[3]

## RESULTS

Numerical results are reported using the mean and standard error of the mean over 5 independent samples. Baseline models are described in Appendix B.

CRYSTAL STRUCTURE IDENTIFICATION

As shown in Figure 2, we select 8 prototypes of single- and two-component crystals from the AFLOW Encyclopedia of Crystallographic Prototypes (Mehl et al., 2017; Hicks et al., 2019). These structures are chosen to demonstrate that models can learn not only geometric information ($cF4$-Cu and $hP2$-Mg are similar structures but with a different stacking of their close-packed layers; the clathrates $cP46$-Si and $cF136$-Si are also similar to each other, with a different arrangement of many common motifs), but also the information encoded within each point ($cP2$-CsCl and $cF8$-ZnS have identical geometry to $cI2$-W and $cF8$-C, respectively, differing only by the type assignments of their particles). For each structure, we rescale the unit cell so that the shortest nearest-neighbor distance over the structure is 1 before replicating the unit cell to consist of at least 2048 particles. We then create three samples of each structure by adding Gaussian noise with a standard deviation of $10^{-3}$, $5 \cdot 10^{-2}$, and 0.1 separately to the particle coordinates, in order to emulate thermal noise. We find the nearest neighbors of each particle using the freud (Ramasubramani et al., 2020) python library and use these point clouds to train models to identify the source structure type for each particle.

These networks rapidly learn to identify structures after a few epochs, with a final overall accuracy of 98.98% ± 0.08% after training for roughly 50 minutes on an NVIDIA Titan Xp GPU. We note that we do not expect this to be a difficult task in general: a baseline method using a rotation-invariant spherical harmonic featurization of varying-sized local neighborhoods of particles (Spellings & Glotzer, 2018) is also able to quickly reach 96.1% ± 0.1% accuracy, although we find encoding particle type information to be more natural in the graph neural network-like setting of the geometric algebra attention networks than when using spherical harmonic featurizations.

MOLECULE FORCE REGRESSION

In a method similar to Batzner et al. (2021), we train models to predict the atomic forces calculated using *ab initio* molecular dynamics and density functional theory available in the MD17 dataset (Chmiela et al., 2017) for eight different molecules. Consistent with previous benchmarks on this dataset, we train networks using the mean squared error loss for each molecule using 1,000 snapshots of forces each as training, validation, and test data sets. Training a model on an individual molecule's data takes between roughly 20 hours (ethanol and malonaldehyde, with nine atoms each) to 90 hours (naphthalene, with eighteen atoms) on an NVIDIA Titan Xp GPU. Test set losses, expressed as the mean absolute error over each force component for each sample, are presented in Table 1.

Table 1: Mean absolute error of force components (in $\frac{meV}{\text{Å}}$) for geometric algebra attention networks, GemNets (Klicpera et al., 2021), NequIP (Batzner et al., 2021), and SchNet (Schütt et al., 2017) architectures.

| Molecule | This work | GemNet-Q | NequIP | SchNet |
|---|---|---|---|---|
| Aspirin | 13.5 ± 0.55 | 9.4 | 15.1 | 58.5 |
| Benzene | 6.68 ± 0.082 | 6.3 | 8.1 | 13.4 |
| Ethanol | 4.6 ± 0.14 | 3.8 | 9.0 | 16.9 |
| Malonaldehyde | 8.3 ± 0.18 | 6.9 | 14.6 | 28.6 |
| Naphthalene | 4.1 ± 0.11 | 2.2 | 4.2 | 25.2 |
| Salicylic acid | 8.0 ± 0.23 | 5.4 | 10.3 | 36.9 |
| Toluene | 3.8 ± 0.17 | 2.6 | 4.4 | 24.7 |
| Uracil | 6.0 ± 0.058 | 4.5 | 7.5 | 24.3 |

Overall, these models perform favorably compared to NequIP and SchNet, while being slightly worse than GemNets. The better performance of GemNets could be due to the operation of GemNet-Q on quadruplets of atoms (the geometric algebra attention networks learned here only operate on pairs of bonds from a central atom), the incorporation of energy in addition to force labels in the GemNet training scheme (in this work, we choose to only train on force data, rather than using a

---

[3]For applications of this method to systems at nonzero temperature, we would expect to be better-served by using an architecture that generates output distributions instead of only point values, but we disregard this here for simplicity; in other words, here we are teaching models to memorize training data directly, rather than model a thermodynamic ensemble.

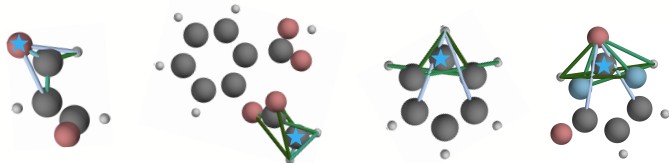

Figure 3: Sample pairwise attention maps for four training data molecules (malonaldehyde, aspirin, benzene, and uracil) after filtering out low-attention pairs. The attention maps indicate how strongly the pair of atoms joined by the line affect the representation of the atom indicated with a star, with lighter lines indicating greater influence. Qualitatively, more complex bonding environments such as those on the right tend to have longer-range attention interactions than the simpler environments on the left.

weighted loss incorporating both quantities), or simply a more suitable architecture or hyperparameters for the GemNet models.

Another useful attribute of attention-based models lies in analyzing learned attention maps (Rogers et al., 2020); to that end, we present sample attention maps of molecules in the training set in Figure 3. Notably, the attention weights tend to be more broadly distributed when calculating the forces on atoms participating in more complex bonding environments—such as aromatic rings—which we may expect due to physical effects, such as electron delocalization.

### PROTEIN COARSE-GRAIN BACKMAPPING

We use 19 protein structures, listed in Appendix C, which have high-resolution structural refinements (with resolution error less than or equal to $1.0\,\text{Å}$) and were published on the Protein Data Bank (Berman et al., 2000) between 2015 and 2020 for training data. Because the variety of structural refinement workflows could lead to a large variance in structure fidelity compared to the learning capacity of the models, we use the training set error to characterize model performance instead of measuring generalization capability with a typical split of training, validation, and test data[4]. We compare geometric algebra attention models to baselines using geometrically-naïve transformer models in order to probe the impact of rotation equivariance on performance. This baseline model does not have rotation equivariance built in, and must learn to approximate it through data augmentation. We test a series of models using representations of varying width to compare the data efficiency of our geometric algebra attention scheme. In addition to comparing the overall accuracy of model predictions, we record the number of epochs required to reach 75% of the total improvement in training error. As shown in Figure 4, equivariant models are typically faster to train and more data-efficient, producing higher-quality predictions using fewer parameters than non-equivariant models. Furthermore, we are able to train larger equivariant models than geometrically-naïve models: basic transformers begin to show difficulty converging to a low training error above 64 neurons, while equivariant models do not saturate in performance until roughly 96 neurons in width.

### DISCUSSION

We find the architectures formulated here to be useful for a variety of tasks. Rather than being limited to operating on bond distances and angles as in SchNet (Schütt et al., 2017), PhysNet (Unke & Meuwly, 2019), and DimeNet (Klicpera et al., 2019), geometric algebra provides a systematic way to build functions with the desired rotation- and permutation-equivariance, with the flexibility to incorporate other types of geometric objects (such as the orientation quaternion commonly used for anisotropic particles in molecular dynamics methods (Kamberaj et al., 2005)) into the framework. The attention mechanism presented here provides a simple, powerful method to account for both geometric and node-level signals. The primitives of our geometric algebra attention scheme—distances,

---

[4]In other applications of this method, we would expect data to be more homogeneous so that typical training, validation, and test data splitting methods can be used; here, we use PDB data as a proxy for simulation outputs using typical simulation methods.

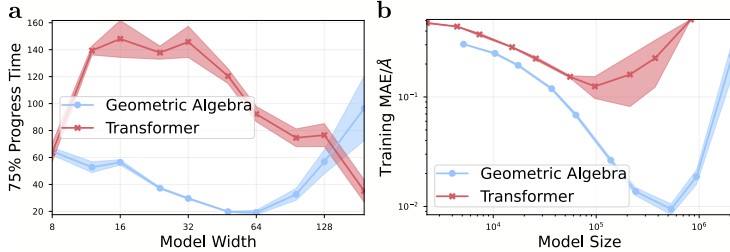

Figure 4: Performance of rotation-equivariant geometric algebra attention networks compared to a naïve transformer-based architecture using dot product attention. Rotation-equivariant models typically improve both (a) the speed of convergence during training and (b) final accuracy obtained.

areas, angles, and volumes—and the calculated attention weights naturally lend themselves to interpretability, which we believe will prove useful in distilling insights from trained models.

Although the architectures presented here work well for the problems we have selected, creating geometric products of vectors is only a subset of the valid combinations that could be generated. In these cases we have carefully chosen sums and differences of input vectors to respect symmetries we would like to impose on the system—such as using the pairwise distance of all input coordinates for the molecular force regression task to impose translation invariance—but it is possible that incorporating learned linear combinations of inputs or even generating intermediate multivector quantities could yield more powerful models.

An obvious limitation to using higher-degree correlations lies in the computational complexity and memory scaling of generating tuples, which are both proportional to $N^r$ for neighborhoods of $N$ coordinates and tuples of length $r$. Polynomial scaling behavior can be ameliorated by restricting which combinations of input points are considered, essentially treating the attention weights of all other combinations as 0. These combinations could be randomly sampled from all valid indices $ijk...$ or use more physically-relevant restrictions, such as utilizing the molecular connectivity graph for molecular force regression or edges derived from the Voronoi tessellation for other applications. If smoothness of model predictions is a concern—as may be the case for learning general N-body interaction potentials, for example—the architectures presented here could be augmented by incorporating weights that decay to 0 as bonds are broken in the Voronoi diagram graph (Mickel et al., 2013).

## CONCLUSION

In this work, we have presented a strategy for developing rotation- and permutation-equivariant neural network architectures by combining geometric algebra and attention mechanisms. These architectures operate directly on vector, scalar, and other geometric quantities of interest to produce outputs which respect desirable symmetries by construction. We believe that the mathematical simplicity and the insights derived from inspecting attention maps are particularly appealing aspects of the algorithms presented here. We hope that these architectures will help a wider range of scientific disciplines reap the benefits of geometric deep learning.

### REPRODUCIBILITY STATEMENT

In addition to the mathematical description of the architectures used in the Methods section, schematic diagrams and pseudocode to generate models for each task in the style of the `tensorflow.keras` API are listed in Appendix B. Descriptions of baseline methods for the structure identification and coarse-grain backmapping tasks are also available in Appendix B. The full code used to train and analyze the new model architectures presented here as well as baselines are available in the Supplementary Information.

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

## A  GEOMETRIC ALGEBRA

The geometric algebra was developed in the 19th century and provides a consistent framework for dealing with scalars and other geometric quantities—such as vectors, areas, and volumes found in three-dimensional space—in arbitrary dimensions (Clifford, 1878). Here, we will describe the essential parts of geometric algebra as related to our proposed attention mechanism, and defer to other works, such as Macdonald (2017), for a more thorough description. The geometric algebra specifies a binary operator, the *geometric product*, that works on *multivectors*. Multivectors can be expressed as linear combinations of terms from a fixed basis set for a given space, such as $\mathbb{R}^2$ or $\mathbb{R}^3$; in three-dimensional space, this yields scalars, vectors, *bivectors* (which specify signed areas within a plane and have 3 components), and *trivectors* (which specify signed volumes and have 1 component)—a total of 8 linearly independent terms for each multivector.[5] When rotation invariance is desired, as in Equation 1, we can utilize the rotation-invariant components of a multivector: scalars, trivectors, the norms of vectors, and the norms of bivectors are rotation-invariant. Geometric algebra provides a general framework that can be used to build up expressive functions as linear combinations and geometric products of multivector inputs; in this work, we derive rotation-equivariant quantities from geometric products to suit the application and symmetry of our problems of interest.

Terms for arbitrary geometric products can be calculated using basic mathematical attributes of the geometric product: for two orthogonal vectors $\vec{e}_1$ and $\vec{e}_2$, $\vec{e}_1\vec{e}_1 = \vec{e}_2\vec{e}_2 = 1$ and $\vec{e}_1\vec{e}_2 = -\vec{e}_2\vec{e}_1$. As an example, multiplying a vector $A = \alpha\vec{e}_1 + \gamma\vec{e}_3$ and vector-bivector combination $B = \delta\vec{e}_1 + \zeta\vec{e}_2 + \mu\vec{e}_1\vec{e}_3 + \nu\vec{e}_2\vec{e}_3$ proceeds as follows:

$$
\begin{aligned}
AB &= (\alpha\vec{e}_1 + \gamma\vec{e}_3)(\delta\vec{e}_1 + \zeta\vec{e}_2 + \mu\vec{e}_1\vec{e}_3 + \nu\vec{e}_2\vec{e}_3) \\
&= \alpha(\delta\vec{e}_1\vec{e}_1 + \zeta\vec{e}_1\vec{e}_2 + \mu\vec{e}_1\vec{e}_1\vec{e}_3 + \nu\vec{e}_1\vec{e}_2\vec{e}_3) + \gamma(\delta\vec{e}_3\vec{e}_1 + \zeta\vec{e}_3\vec{e}_2 + \mu\vec{e}_3\vec{e}_1\vec{e}_3 + \nu\vec{e}_3\vec{e}_2\vec{e}_3) \\
&= \alpha(\delta + \zeta\vec{e}_1\vec{e}_2 + \mu\vec{e}_3 + \nu\vec{e}_1\vec{e}_2\vec{e}_3) + \gamma(-\delta\vec{e}_1\vec{e}_3 - \zeta\vec{e}_2\vec{e}_3 - \mu\vec{e}_1\vec{e}_3\vec{e}_3 - \nu\vec{e}_2\vec{e}_3\vec{e}_3) \\
&= \alpha(\delta + \zeta\vec{e}_1\vec{e}_2 + \mu\vec{e}_3 + \nu\vec{e}_1\vec{e}_2\vec{e}_3) + \gamma(-\delta\vec{e}_1\vec{e}_3 - \zeta\vec{e}_2\vec{e}_3 - \mu\vec{e}_1 - \nu\vec{e}_2) \\
&= \underbrace{\alpha\delta}_{\text{scalar } \alpha\delta} \underbrace{-\gamma\mu\vec{e}_1 - \gamma\nu\vec{e}_2 + \alpha\mu\vec{e}_3}_{\text{vector } (-\gamma\mu, -\gamma\nu, \alpha\mu)} + \underbrace{\alpha\zeta\vec{e}_1\vec{e}_2 - \gamma\delta\vec{e}_1\vec{e}_3 - \gamma\zeta\vec{e}_2\vec{e}_3}_{\text{bivector } (\alpha\zeta, -\gamma\delta, -\gamma\zeta)} + \underbrace{\alpha\nu\vec{e}_1\vec{e}_2\vec{e}_3}_{\text{trivector } \alpha\nu}
\end{aligned}
\tag{4}
$$

More generally, the types of elements produced by a geometric product of two multivectors in $\mathbb{R}^3$ with the given components are summarized in Table 2 below.

Table 2: Terms arising from the geometric product $AB = (A_s + A_v + A_b + A_t)(B_s + B_v + B_b + B_t)$ in $\mathbb{R}^3$. In three dimensions, multivectors $A$ and $B$ consist of scalars ($s$), vectors ($v$), bivectors ($b$), and trivectors ($t$).

|           | $\mathbf{B_s}$ | $\mathbf{B_v}$ | $\mathbf{B_b}$ | $\mathbf{B_t}$ |
|-----------|------|---------|---------|------|
| $\mathbf{A_s}$ | s    | v       | b       | t    |
| $\mathbf{A_v}$ | v    | s + b   | v + t   | b    |
| $\mathbf{A_b}$ | b    | v + t   | s + b   | v    |
| $\mathbf{A_t}$ | t    | b       | v       | s    |

From Table 2, we can see that successive products of vectors—such as $p_{ijk...}$ in Equation 1—alternate between producing two types of multivectors: products of even numbers of vectors yield a scalar and bivector $((v + t)v = vv + tv = (s + b) + b \to s + b)$, while products of odd numbers of vectors produce a vector and trivector $((s + b)v = sv + bv = v + (v + t) \to v + t)$. Generating rotation-invariant quantities from these products as described in the main text is the primary application of geometric algebra in this work, although in general the method could be used to incorporate different types of scalar, vector, bivector, and trivector quantities; for example, rotations could be input as quaternions, which are isomorphic to the scalar-and-bivector product of even numbers of vectors.

---

[5]Multivectors form a vector space: the individual components of multivectors (bivectors, for example) can be directly summed elementwise, but multivector components of different types stay separate and are multiplied using the distributive property of geometric products when needed, producing output types according to Table 2. The geometric product has an identity of the scalar 1 and is associative; in other words, it forms a monoid over multivectors.

ROTATION EQUIVARIANCE

Because the scalars used in Equation 2 are rotation-invariant and geometric algebra is coordinate-free—that is, we can perform a change of basis to replace the basis vectors $\vec{e}_1$, $\vec{e}_2$, $\vec{e}_3$ with rotated versions $\mathbf{Q}\vec{e}_1$, $\mathbf{Q}\vec{e}_2$, $\mathbf{Q}\vec{e}_3$ to transform the point cloud using a rotation matrix $\mathbf{Q}$ and the vector component of the product $p_{ijk...}$ will transform in the same way—the output of Equation 2 is also rotation-covariant: for a rotation matrix $\mathbf{Q}$,

$$
\begin{aligned}
\mathbf{Q}\vec{r}_i' &= \mathbf{Q}\sum_{jk...} w_{ijk...}\mathcal{R}(v_{ijk...})\left(\alpha_0\text{vector}(p_{ijk...}) + \alpha_1\vec{r}_i + \alpha_2\vec{r}_j + \alpha_3\vec{r}_k + ...\right) \\
&= \sum_{jk...} w_{ijk...}\mathcal{R}(v_{ijk...})\left(\alpha_0\mathbf{Q}\text{vector}(p_{ijk...}) + \alpha_1\mathbf{Q}\vec{r}_i + \alpha_2\mathbf{Q}\vec{r}_j + \alpha_3\mathbf{Q}\vec{r}_k + ...\right) \\
&= \sum_{jk...} w_{ijk...}\mathcal{R}(v_{ijk...})\left(\alpha_0\text{vector}((\mathbf{Q}\vec{r}_i)(\mathbf{Q}\vec{r}_j)(\mathbf{Q}\vec{r}_k)...) + \alpha_1\mathbf{Q}\vec{r}_i + \alpha_2\mathbf{Q}\vec{r}_j + \alpha_3\mathbf{Q}\vec{r}_k + ...\right).
\end{aligned}
$$

## B  MODEL DETAILS

We list python-style pseudocode—using the semantics of the Keras API within Tensorflow (Tensor-Flow Developers, 2021)—for the models used in this work below. We also use the following geometric algebra attention layers, which are built with MLPs given by the listed `make_scorefun` ($\mathcal{S}$) and `make_valuefun` ($\mathcal{V}$) functions described in Equation 1:

- `VectorAttention` implements the value-generating geometric algebra attention calculation described in Equation 1. It accepts a point cloud and set of values associated with each point, and produces new values.
- `Vector2VectorAttention` implements the coordinate-generating (i.e., rotation-covariant) geometric algebra attention calculation described in Equation 2. It accepts a point cloud and set of values associated with each point, and produces new coordinates.
- `LabeledVectorAttention` Implements the label-augmented point cloud translation layer described in Equation 3. It takes three arguments (a set of label values, a set of reference coordinates, and a set of reference values) and produces a set of new coordinates.

Listing 1: Sample model-building code for rotation-invariant crystal structure identification.

```
def make_scorefun():
    return Sequential([Dense(64, activation='relu'),
                       Dropout(0.5), Dense(1)])

def make_valuefun():
    return Sequential([Dense(64), LayerNormalization(),
                       Activation('relu'), Dropout(0.5),
                       Dense(32)])

# Inputs are given as N-length sets of coordinates
# or values for each nearest-neighbor bond
r_in = Input((None, 3))
v_in = Input((None, D_in))
# Project values to the desired working dimension
last_v = Dense(D_working)(v_in)

# Calculate attention in a series of blocks
for _ in range(2):
    residual_in = last_v
    last_v = VectorAttention()(r_in, last_v)
    last_v = Dense(2*D_working, activation='relu')(last_v)
```

```
    last_v = Dense(D_working)(last_v) + residual_in

# Permutation-invariant attention summation (NxD_working)
last_v = VectorAttention(permutation_invariant=True)(r_ij, last_v)
# Final MLP
last_v = Dense(2*D_working, activation='relu')(last_v)
logits = Dense(num_classes, activation='softmax')(last_v)
model = Model([r_in, v_in], logits)
```

Listing 2: Sample model-building code for conservative, rotation-equivariant neural networks for molecular force regression.

```
def make_scorefun():
    return Sequential([Dense(64, activation='swish'),
                       Dense(1)])

def make_valuefun():
    return Sequential([Dense(64), LayerNormalization(),
                       Activation('relu'), Dense(32)])

# Inputs are given as N-length sets of coordinates
# or values for each atom in a molecule
r_in = Input((None, 3))
v_in = Input((None, D_in))
# Convert to atom-centered coordinates (NxNx3) and values (NxNx[2*D_in])
r_ij = PairwiseDifference()(r_in)
v_ij = PairwiseSumDifference()(v_in)
# Project values to the desired working dimension
last_v = Dense(D_working)(v_ij)

# Calculate attention in a series of blocks
for _ in range(6):
    residual_in = last_v
    last_v = VectorAttention()(r_ij, last_v)
    last_v = Dense(2*D_working, activation='swish')(last_v)
    last_v = Dense(D_working)(last_v) + residual_in

# Permutation-invariant attention summation (NxD_working)
last_v = VectorAttention(permutation_invariant=True)(r_ij, last_v)
# Final MLP and summation
last_v = Dense(2*D_working, activation='swish')(last_v)
atomic_energy = Dense(1, use_bias=False)(last_v)
# The energy of the molecule is the sum of each atom's energy
molecule_energy = NeighborhoodSum()(atomic_energy)
# The force is the negative gradient of the energy
# with respect to input coordinates
force_output = NegativeGradient()(molecule_energy, r_in)
model = Model([r_in, v_in], force_output)
```

Listing 3: Sample model-building code for rotation-covariant coarse-grain operator backmapping.

```
def make_scorefun():
    return Sequential([Dense(64, activation='relu'),
                       Dense(1)])

def make_valuefun():
    return Sequential([Dense(64), LayerNormalization(),
                       Activation('relu'), Dense(32)])
```

```
# Inputs are given as N-length sets of coordinates
# or values for each coarse-grained neighbor, and a set
# of atomic labels for the central coarse-grained bead
r_in = Input((None, 3))
v_in = Input((None, D_in))
child_types_in = Input((None,))
# Project values to the desired working dimension
child_labels = Embedding(num_child_types)(child_types_in)
last_v = Dense(D_working)(v_in)

# Calculate attention in a series of blocks
for _ in range(2):
    residual_in = last_v
    last_v = VectorAttention()(r_in, last_v)
    last_v = Dense(2*D_working, activation='relu')(last_v)
    last_v = Dense(D_working)(last_v) + residual_in

# Translate coarse-grained to fine-grained coordinates
vecs = LabeledVectorAttention()(r_in, last_v, child_labels)

# Convert to atom-centered representations for refinement
delta_v = PairwiseDifferenceSum()(child_labels)
delta_v = Dense(D_working)(delta_v)

# Refine atomic coordinates
for _ in range(2):
    residual_in = vecs
    vecs = PairwiseDifference()(vecs)
    vecs = Vector2VectorAttention(permutation_invariant=True)(
        vecs, delta_v)
model = Model([r_in, v_in, child_types_in], vecs)
```

We also present diagrams summarizing the geometric algebra attention model architectures used in this work in Figure 5.

BASELINE MODELS

**Crystal structure identification.** We train MLPs on spherical harmonic descriptions of varying-sized local environments of particles as described in Spellings & Glotzer (2018). To deal with particle types using this method, we concatenate two sets of local environment spherical harmonics: one considering only bonds of like-typed particles (i.e. bonds to particles of type A if the central particle is of type A), and another set considering only bonds to differently-typed particles. We consider environments of up to the 12 nearest neighbors (disregarding particle types) for each set of spherical harmonics. The MLPs used consist of a batch normalization layer followed by two blocks of a dense layer 64 neurons in width using a ReLU activation function feeding into a dropout layer with a dropout rate of 0.5. The final result is projected down to class logits.

**Backmapping of coarse-graining operators.** We train models using dot product attention: vertex representations are concatenated directly with their geometric coordinates and fed into a similar architecture to that shown in Figure 5, with geometric algebra attention layers replaced by standard dot-product attention layers. For the intermediate translation layer, query embeddings associated with each atom type are used. Because this baseline model is not rotation-covariant by construction, it must learn to generate rotation-covariant predictions; to impart this information to the model, we apply a random rotation to each pair of point clouds during training.

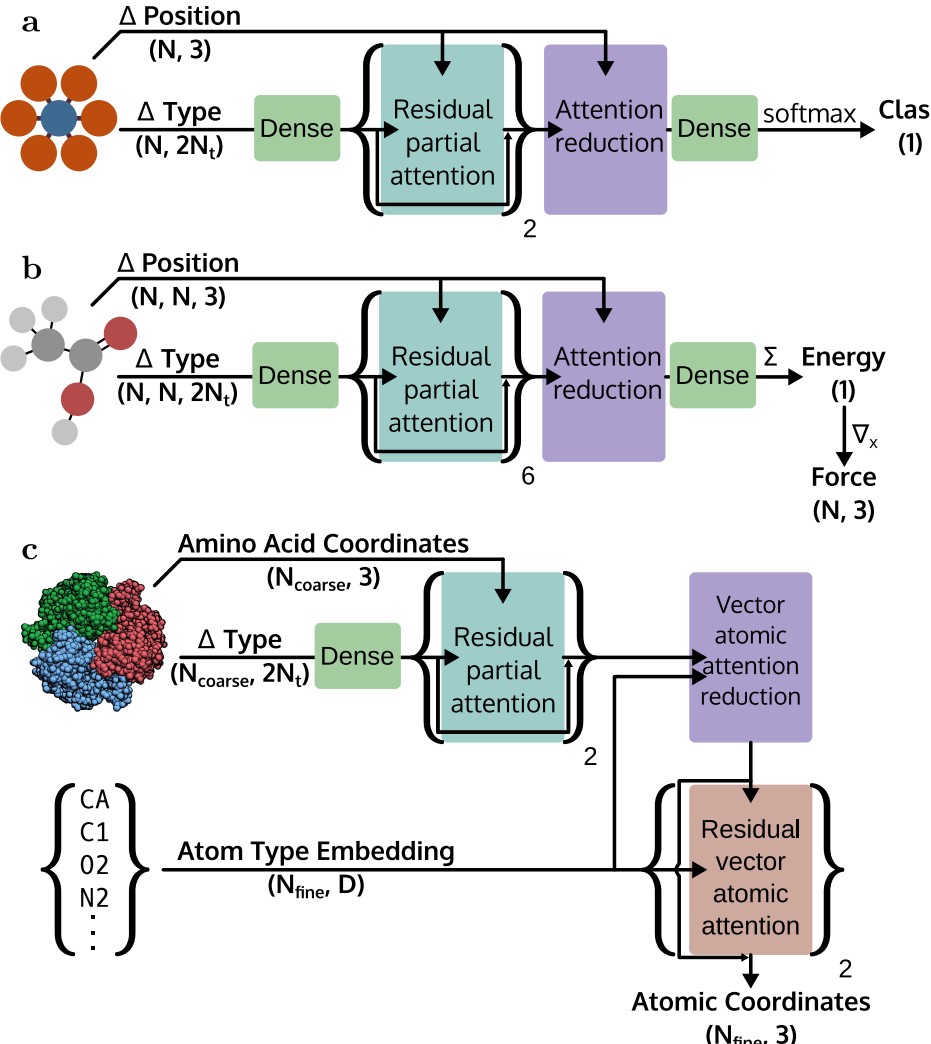

Figure 5: Visual overview of model architectures used in this work. Architectures presented are for (a) crystal structure classification, (b) molecular force regression, and (c) coarse-grained backmapping.

MODEL TRAINING

Networks for the crystal structure identification and protein coarse-grain backmapping tasks are trained for up to 800 epochs using the adam optimizer (Kingma & Ba, 2015); the learning rate is decreased by a factor of 0.75 after the validation set loss does not decrease for 20 epochs, and training is ended early if the validation set loss does not decrease for 50 epochs. Models trained on the molecular force regression task use the longer training schedule presented in Batzner *et al.*, which decreases the learning rate by a factor of 0.8 after 1000 epochs of no improvement in the validation loss and ends training after 2500 epochs of no improvement, or 50000 epochs in total.

## C PDB ENTRIES USED FOR COARSE-GRAINING TASK

- 3X2L (Nakamura et al., 2015)
- 3X32 (Hirano et al., 2015)
- 4TKJ (Matsuoka et al., 2015)
- 3X0J (Furuike et al., 2016)

- 5WQR (Ohno et al., 2017)
- 6EQE (Austin et al., 2018)
- 6ETK (Vergara et al., 2018)
- 6FJN (Baker et al., 2019)
- 6JGJ (Takaba et al., 2019)
- 6RYG (Paterson et al., 2019)
- 6IIP (Tian et al., 2019)
- 6Q01 (Schellenberg et al., 2020)
- 6XVM (Plaza-Garrido et al., 2020)
- 6Y5S (Wu et al., 2020)
- 6YP6 (Franke et al., 2020)
- 7A5M (Barone et al., 2020)
- 7K4T (Otten et al., 2020)
- 6YK4 (Frydenvang et al., 2020)
- 6SAY (Glöckner et al., 2020)

## D  MODEL EVALUATION THROUGHPUT MEASUREMENTS

Typical times required to evaluate models on a single point cloud using an NVIDIA GeForce RTX 2060 GPU, amortized over a batch, are reported in Table 3 below.

Table 3: Times required to evaluate models on point clouds.

| Task | Model details | Time (ms/point cloud) |
| --- | --- | --- |
| Crystal Structure Identification | | 0.19 |
| Molecular Force Regression | Aspirin | 7.6 |
| | Benzene | 4.2 |
| | Ethanol | 5.1 |
| | Malonaldehyde | 5.5 |
| | Naphthalene | 6.5 |
| | Salicylic Acid | 6.1 |
| | Toluene | 5.3 |
| | Uracil | 4.4 |
| Coarse-grain Backmapping | 8–128 neurons | 0.67 |
| | 192 neurons | 1.0 |

