# OpenReview forum: "Geometric Algebra Attention Networks for Small Point Clouds"
_ICLR.cc/2022/Conference — ICLR 2022 Submitted_

### Official Review · Reviewer_sEBH · 2021-10-27

**Correctness:** 2
**Technical Novelty And Significance:** 3
**Empirical Novelty And Significance:** 3
**Recommendation:** 3
**Confidence:** 4

**Main Review:**

Strengths:
+ It is interesting that the idea of utilizing the theory of geometric algebra to achieve equivariance.
+ The demonstration broadly covers sample problems relevant to physics, chemistry, and biology.

Weaknesses:
- The authors do not clearly describe how the geometric product helps to achieve the rotation-invariant attributes. The authors should explain it in the main manuscript since this is one of the main contributions.
- Geometric products do not appear in both model architectures and appendix B.  The authors are required to exlicitly describe how it is implemented.
- I think the interpretation of the linear combination in geometric algebra is not trivial. For example, how can we interpret the average of a point and a plane, which are commonly expressed as geometric numbers? Even for the point, a linear combination of some points would be interpreted as a line when the scalar part is 0 (~ eq(2)). Is this appropriate? This paper lacks the interpretation, justification, and discussion about the attention mechanism in geometric algebra.
- Why is the proposed architecture limited to "small point clouds"? If the statement about the computational complexity is the reason, the connection to it should be clarified.

Minor comments:
- I supposed that this paper uses the projective geometric algebra Cl(3,0,1), but not clarified. Even for 3D, the conformal geometric algebra Cl(4,1,0) might be a candidate. Clarification, justification, and discussion are appreciated.
- The authors defined four functions, V, M, J, and S. However, another function R is defined later. Why do the authors distinguish R from the others? Are the four functions cover everything sufficiently? Justification is required.
- y_i in eq(1) seems output, but the statement "a network producing permutation-covariant outputs for each input point y_i" mislead y_i to be input point.
- Is the geometric product commutative? If not, how do the authors achieve permutation-covariance for varying p?
- Is "working width"  in MODEL ARCHITECTURES "working dimension"?
- Is it possible to perform molecule force regression for unknown molecule?
- The authors stated that GemNets can operate on quadruplets of atoms but the proposed network worked only in pairs of bonds from the central atom. If this is the limitation of the proposed network, clear statement and discussion are required. Moreover, a discussion is appreciated whether the proposed network has technical difficulties to include the force label.
- I could not find which point "show difficulty in convergence above 64 neurons" in protein coarse-grain backmapping.



**Summary Of The Paper:**

This paper presents a deep neural network for processing small sets of points in three-dimensional space with a rotation-invariance and a permutation equivariance.
The proposed method is twofold. One utilizes a geometric product of geometric algebra on input vectors to achieve the rotation-invariant attributes. The other is a permutation-equivariant reduction over the geometric products using an attention mechanism.
The authors demonstrate applications in physical science such as crystal structure identification, molecular force regression, and backmapping of coarse-graining operators.


**Summary Of The Review:**

This manuscript has several issues to be accepted as raised in the weaknesses.

---

> ### Author Response · Authors · 2021-11-18
> **Initial author response**
>
> We thank reviewer sEBH for their comments and will strive to address them here and with revisions in the text as follows:
>
> In response to “Weaknesses:”
>
> * We will describe in more detail how the geometric product helps incorporate rotation-invariant geometric information into the attention mechanism.
> * We have attempted to clarify what we mean by the geometric product of vectors $r_i r_j$ in the text, but could the reviewer confirm that this is the type of information they were suggesting? We tried to make the geometric products of input vectors explicit as the $p_{ijk…}$ terms of the attention calculation (equation 1) and step through what a geometric product calculation looks like in Appendix A (equation 4).
> * In the current work, we do not perform linear combinations of multivector quantities; we only use the attention mechanism to generate node-level signals or vectors, which we believe are more straightforward to understand in terms of linear combinations. We believe that combinations of multivector quantities could be an interesting idea to pursue as a more powerful way to incorporate geometry into the calculations of the network (as the networks presented here end up using the same multivectors of the input points at every layer, assuming pairwise attention for each layer), but we leave that idea for future work. Perhaps the approach presented here may not be the most general formulation of using geometric algebra to incorporate geometric information into an attention mechanism, but we believe that its simplicity is valuable and that this is an interesting avenue of research to pursue. We will add discussion on these points, but could the reviewer clarify which parts of the attention mechanism could use more interpretation, justification, and discussion?
> * The main issue limiting the method to small point clouds is the memory size of exhaustively calculating the combinations of input points. A more clever algorithm could iterate over these combinations sequentially, but the computational complexity would eventually limit the efficiency of the method. We will add some ideas to deal with these limitations to the discussion section.
>
> In response to “Minor comments:”
>
> * The authors are not familiar with the taxonomy of Clifford algebras, having learned of geometric algebra primarily from the work of Alan Macdonald. Looking into other literature, we believe that this point brings up an interesting area of further study! We thank the reviewer for this insight and intend to improve our knowledge in this area.
> * Without the scaling factor $\mathcal{R}$ or the combination terms $\alpha_i$, the output vector is restricted to be a convex combination of the input vectors for a given point cloud. The $\alpha_i$ terms are learned but are not functions of the values or geometry of the point cloud, so $\mathcal{R}$ gives additional freedom to rescale the output vector based on geometry and node-level signals. There are certainly many ways to generate rotation-covariant vectors from attention schemes like the one presented here, so we chose a fairly simple factorization, but one could imagine several other variants. We will elaborate in the text on this topic.
> * We had originally misplaced the $y_i$ term in that sentence; we have updated the text accordingly.
> * The geometric product is not commutative; we attain permutation equivariance by summing over all (pairwise, triplet-wise, and so on) combinations of input points in a cloud.
> * We did indeed mean “working dimension” rather than “working width;” we have updated the text to use more consistent terminology.
> * One of the nice aspects of using an attention mechanism is the ability to deal with molecules of varying size. To that extent, the models we train here could be used for new, unknown molecules. Predictions from a network trained on one set of molecules could conceivably be accurate on a new molecule that has not been seen before, given that the functions learned are all atom-centered.
> * Using quadruplets of bonds as in GemNets is only difficult to implement in our attention mechanism because we exhaustively calculate all combinations of bonds for a given attention rank, which scales poorly in the size of the input point clouds. There are other strategies that could be used to restrict which combinations of vectors are used in the attention calculation, including making use of the bond graph for molecular systems as GemNets do. We will add more discussion about future strategies to improve the performance of the geometric algebra attention mechanism.
> * By difficulty in convergence, we mean that the networks do not reliably train well to a low-error configuration. We will try to clarify our meaning in the text.

---

> > ### Comment · Reviewer_sEBH · 2021-11-19
> > **Answer to the questions**
> >
> > I would thank the authors' response. Let me clarify some points asked by the authors.
> >
> > > could the reviewer confirm that this is the type of information they were suggesting?
> >
> > I meant I could not find a connection from theory (eq1 and App. A) to implementation ("Model architecture" section and App.B).
> >
> > > could the reviewer clarify which parts of the attention mechanism could use more interpretation, justification, and discussion?
> >
> > As the authors stated, geometric algebra provides mathematical structure to deal with geometric objects. I think this property is maintained under any (not only geometric) "products", but I'm not sure under "sum" (linear combination).
> > I would ask the authors to add the discussion about geometrical interpretation about this point.

---

> > > ### Author Response · Authors · 2021-11-19
> > > **Follow-up response to clarifications**
> > >
> > > We thank the reviewer for their clarification. In addition to the notes on the geometric interpretation of the multivector intermediates used in the network that were added in the first revision, the next revision we post will include explicit links to the equations described in the Methods section for each network architecture section.

---

### Official Review · Reviewer_6SEn · 2021-11-01

**Correctness:** 4
**Technical Novelty And Significance:** 3
**Empirical Novelty And Significance:** 3
**Recommendation:** 6
**Confidence:** 2

**Main Review:**

# Strengths:
- Formulation of rotation-equivariance by leveraging generic geometric products.
- New attention formulation for achieving permutation-equivariance which is important for many deep learning applications in different fields.
- Performance is evaluated in three applications with different datasets which shows the robustness of the model proposed. The evaluation also showcases the importance of geometric properties in deep learning models for data efficiency, robust prediction, and fast training.

# weaknesses:
 - In table 1, the proposed model performs worse than the baseline GemNet-Q and some possible reasons for that are enumerated by the authors like the use of quadruplets atoms, the incorporation of energy, or even better a better architecture. Can the proposed method be adjusted to have these properties in order to perform a fair comparison with this baseline? However, the results are good and it is fine to present worse results than some baseline finetuned to the application tested.
 - I did not understand footnote 4. Why is the training error used to evaluate the performance instead of the test error in the protein coarse-grain backmapping? How does this measure the generalization performance of the model?



**Summary Of The Paper:**

This paper proposes a rotation and permutation equivariant geometric deep learning model for problems where the data is represented as small points clouds. The equivariance properties are achieved by leveraging geometric algebra formulations. More specifically, rotation-equivariance is accomplished by geometric products of multivectors and permutation-equivariance by using an attention mechanism over invariant terms of these products. This model is evaluated in three different applications showing better or comparable results than existing approaches. These models also offer additional features like the analysis of the attention maps produced.

**Summary Of The Review:**

I am not an expert on this topic. But, I think this paper proposes a novel formulation to provide rotation and permutation equivariance for deep learning models for small point clouds which are useful in many applications. They also show improvements over existing approaches in three different applications. Therefore, I suggest accepting this paper.

---

> ### Author Response · Authors · 2021-11-18
> **Initial author response**
>
> We thank reviewer 6SEn for their feedback and comments. Our responses are below:
>
> * While we could in principle use rank-3 attention for triplets of bonds from a central atom for the molecular force regression task, the current implementation—which would exhaustively calculate all triplets of bonds from each central atom, in this case—would be difficult to fit on current GPUs due to memory limitations. This is particularly true for the molecular force regression task due to making the calculations atom-centered, so for triplet-level attention the complexity would actually scale as the molecule size to the fourth power. The GemNet architecture is able to restrict its quadruplets to those atoms connected in the molecular graph, which makes the calculation much simpler. Our method could be adapted to incorporate information from the molecular graph—either by augmenting the signal for each bond to incorporate bond type information, or reducing the complexity by only calculating attention for bonded atoms—but, for this work, we decided to start with the simplest, naive approach. With this feedback in mind, we will add discussion on the topic of restricting attention to neighbors in the molecular graph to the manuscript. On the topic of incorporating energy as well as force into the loss, this is trivial from a technical perspective, but introduces another hyperparameter controlling the balance of the force and energy terms which we decided to excise for simplicity.
> * The main issues with the coarse-grain backmapping task as presented here come from using entries from the PDB to generate training data. Because generating each PDB entry uses its own experimental protocol—likely including energy minimization using fitted interactions for final coordinates—there are many hidden variables involved in these experiments and the uncertainties in the calculations are larger or of comparable size to the errors learnable by our models. For this reason, we believe that it is difficult to generalize with low error on the dataset we use for this task. Actual applications of this method would have users generate their own data with a coarse-grained model in more uniform conditions and at nonzero temperature. In that setting, it would be much more straightforward to perform a typical train-test split of the data. We will try to clarify this topic in the main text.

---

### Official Review · Reviewer_TD8X · 2021-11-02

**Correctness:** 4
**Technical Novelty And Significance:** 3
**Empirical Novelty And Significance:** 4
**Recommendation:** 6
**Confidence:** 3

**Main Review:**

Rotation equivariance based on geometric algebra is novel and impressive. The proposed attention also shows the impressive performance on several real-world tasks.

However, I still have several concerns which I will detail below

- The details of the four functions are a little vague to me. For example, what are rotation-invariant geometric quantities for tuples? I recommend elaborating more on the technical details if possible to improve readability.

- Mathematical proof of rotational equivariance is missing.

- While the paper shows an impressive performance in real-world tasks, I wonder about the efficiency in particular compared with other works.

- Missing rotation equivariance experiments are missing. I recommend having an experimental test for the rotation equivariance by comparing with other works -- say rotation equivariance achieved via data augmentation.

- Is it possible to extend work into relatively larger point clouds? Also, I am actually not quite sure what the definition of small point clouds is. Is that determined by the size of point clouds? If so, what’s the threshold then? I would recommend having a simple experiment on 3D point cloud classification -- e.g., the popularly used Modelnet 40. The number of points could be controlled via sampling.


**Summary Of The Paper:**

The paper proposes a geometric algebra attention network for small point clouds. The attention based on geometric algebra’s multivector is rotation equivariant and permutation equivariant. Specifically, attention is composed of four functions that operate on tuples and respect the desired equivariance. Moreover, the paper validates its proposed geometric algebra attention on three domain-specific applications including crystal structure identification, molecular force regression, and back mapping of coarse-graining operators.


**Summary Of The Review:**

I am overall impressed by the idea of geometrical algebra-based rotation equivariance. The experimental results are also impressive. However, from my viewpoint, the paper still has several weaknesses in terms of clarity and experimental validation. I would like to hear more during the rebuttal phase. Thus, currently, I vote for weakly accept.

---

> ### Author Response · Authors · 2021-11-18
> **Initial author response**
>
> We thank reviewer TD8X for their insightful comments and will strive to address the concerns in the text as listed below:
>
> * The rotation-invariant geometric quantities associated with tuples depend on the rank of the calculation—that is, the number of vectors producing the geometric product. For a single vector (although rank 1 attention is not used in the examples, since it only makes sense for permutation-invariant reductions), this is just that vector’s magnitude; for even numbers of vectors, this is the scalar component and magnitude of the bivector component. For odd numbers of vectors, it is the magnitude of the vector component and the value of the trivector component. We will try to make this link more clear in the text.
> * We have added a discussion around rotation equivariance in the main text and appendix.
> * We have added a table to Appendix D giving evaluation time for various models trained in this work. One point of comparison is that our model takes around 5 ms to evaluate the force for toluene, compared to 16 ms for the NequIP work.
> * Just to emphasize, we do compare the performance of equivariant models to non-equivariant models with data augmentation in the protein coarse-grain backmapping task; would the reviewer say that we need to compare to non-equivariant models for all tasks? We agree that it could be interesting to compare as a baseline, but it would be difficult to run so many new experiments in a short period of time.
> * We emphasize that these are small point clouds simply due to the polynomial scaling of the calculation and limited memory on modern-day GPUs; this restricts the current implementation (that naively calculates all pairs, triplets, or so on simultaneously) to something like dozens of points. More clever implementations could sequentially operate on these tuples and work with much larger point clouds, although the overall computational complexity would remain the same. On the topic of ModelNet40, we have completed some preliminary evaluation of models by randomly sampling 8 to 32 points from the meshes and varying the rank of the attention calculation. We feel that it may be best to reserve work on this dataset for the future, as the hyperparameters we used to train most other networks caused the rank 3 attention to perform significantly worse than rank 2 attention for this dataset.

---

### Official Review · Reviewer_vhG8 · 2021-11-02

**Correctness:** 3
**Technical Novelty And Significance:** 3
**Empirical Novelty And Significance:** 2
**Recommendation:** 6
**Confidence:** 3

**Main Review:**

Strengths:
 - The proposed method is principled, general, and convenient. It applies to interactions of arbitrary rank.#
 - The method is described in a detailed and reproducible way.
 - The paper is well written and easy to follow.
 - The setting of the paper is clearly defined as learning functions on small point clouds, where there is an emphasis on higher order interactions and respecting symmetries. The chosen evaluation domains form a diverse set of instantiations of this setting.

Weaknesses:
 - The quantitative results are good, but not revolutionary. The model does well on the crystal classification task, but this setting is described as 'not difficult'. It performs slightly worse than GemNet on molecular force regression. On the backmapping task, it is only compared against a naive transformer, as opposed to another rotation invariant model. Another instance of the proposed model outperforming a strong baseline on a difficult task would significantly improve the case the paper is making.
 - While one of the advantages of the proposed method is scaling to arbitrary rank, the experiments are all limited to pairwise attention (not counting the current key element). It would be interesting to demonstrate the effect of scaling up the rank, especially as this is noted as a potential advantage of the GemNet baseline.
 - Group representation-based approaches have not been compared against.

**Summary Of The Paper:**

The paper proposes a method for learning functions that take small point clouds as input, such that they are both permutation and rotation equivariant. To aggregate information from a tuple of points in a rotation equivariant way, it utilizes the geometric product of the points' 3D coordinates. Permutation equivariance is guaranteed by the standard attention framework.
The model is evaluated on three scientific tasks: Crystal structure identification, Molecular force regression, and backmapping of coarse-grained operators in molecule simulations.

**Summary Of The Review:**

Overall, the paper proposes a convenient and principled method to an interesting problem. The empirical evaluation is generally convincing, but could be somewhat more thorough in its analysis, especially given that the paper is a full page under the page limit.

---

> ### Author Response · Authors · 2021-11-18
> **Initial author response**
>
> We thank Reviewer vhG8 for their helpful comments and suggestions.
>
> * While we agree that more comparisons to baselines would be helpful, it may be difficult to find great baselines for the structure identification and coarse-grain backmapping tasks, as these are not yet as well-established or defined as clearly as tasks typically are in computer science research. To the authors’ knowledge, this is the first application of deep learning to the structure identification problem for complex periodic structures, although the structure identification problem is closely related to point cloud classification as would be used for LIDAR point clouds, for example. Similarly, prior work on coarse-grain backmapping either used MLPs or coerced the problem into that of image translation, which violate permutation and rotation symmetries, respectively. While we could potentially adapt tensor field networks, SE(3) transformers, or other architectures that have been released within the last couple of years to these domains for our own baseline comparison purposes, it would likely be unclear that we are comparing fully optimized hyperparameters for both sets of models. For the molecular force regression task, we note that there is a rich history of work in this setting and that the geometric algebra attention networks compare quite favorably to almost all architectures that we are aware of, except for the GemNets.
> * We agree that increasing the rank would be interesting to probe for these experiments! Unfortunately, due to the naive implementation of the architectures at the moment, it is difficult to get the calculations to comfortably fit within GPU memory. This is especially true for the molecular force regression task due to the atom-centering step, that already makes the calculation scale cubically for pairwise attention, rather than as the square of the molecule size. We have attempted some experiments to test the feasibility of rank-3 attention on the tasks, but we believe that these experiments will require more powerful hardware or careful tuning of hyperparameters to be suitable for publication.
> * We do compare our method to tensor field networks, which we consider to be based on group representations, for the molecule force regression task. Tensor field networks could also be used for the other two tasks, but they have an impressive set of hyperparameters that may be difficult to tune for comparison purposes.

---

> > ### Comment · Reviewer_vhG8 · 2021-11-29
> > **Response**
> >
> > Thank you for your answers and clarifications. I understand the difficulties of obtaining additional baselines and think you chose a reasonable set of experiments. It would be good to establish a set of benchmarks for the "small point cloud" setting in the future, as I think this is a meaningful category. Results using higher rank would be nice, but I think the work also meets the bar for publication as is. I will therefore leave my score at 6.

---

> > > ### Author Response · Authors · 2021-11-29
> > > **Response**
> > >
> > > We thank the reviewer for their feedback and suggestions during the review process. We agree that additional benchmarks for these types of applications would be great; the chemistry community has really worked hard in this area over the last several years and of course there are the CASP challenges, but we agree that many areas in basic physics and materials science could benefit from more readily-available standardized datasets and benchmark tasks.

---

### Decision · Program_Chairs · 2022-01-20

**Decision:**

Reject

**Comment:**

This work studies the problem of building powerful representations of low-dimensional point clouds with permutation and rotational equivariance, with the motivation to tackle applications in the physical sciences. Their main technical contribution is the use of the so-called geometric algebra, a series of operations between scalar and vector quantities that respect rotational symmetries, which the authors then combine with attention mechanisms to provide permutation symmetry.

Reviewers generally found this work full of interesting ideas, in particular the novel geometric algebra structure to deal with rotational symmetry. However, they also found several issues, such as lack of clarity and somewhat unclear experimental validation. In particular, the authors are encouraged to formalise the rotational equivariance property, and to further address the "small" aspect of the title. Taking all these considerations into account, the AC recommends rejection at this time, but encourages the authors to pursue this exciting line of research.